

# Pramipexole has a neuroprotective effect in spinal cord injury and upregulates D2 receptor expression in the injured spinal cord tissue in rats

Xuchen Liu, Chengqiang Wang, Qingshan Peng, Birong Peng and Lixin Zhu

Department of Spinal Surgery, Orthopedic Medical Center, Zhujiang Hospital, Southern Medical University, Guangzhou, China

## ABSTRACT

Spinal cord injury (SCI) has emerged as a prevalent condition with limited effective treatment options. The neuroprotective role of pramipexole (PPX) in inhibiting nerve cell apoptosis in central nervous system injuries is well established. Therefore, we investigated the effects of PPX in SCI. Adult Sprague-Dawley rats were divided into four groups (sham, SCI, PPX-0.25, and PPX-2.0 groups) according to the PPX therapy ($n = 24$). Then, SCI was induced using the modified Allen method, and PPX was intravenously administered into the tail at dosages of 0.25 or 2.0 mg/kg following the injury. Motor function was evaluated using the Rivlin-modified inclined plate apparatus and the Basso Beattie Bresnahan (BBB) workout scale. Western blotting assay was used to measure protein expression levels of DRD2, NeuN, Bax/Bcl-2, and caspase-3. Furthermore, immunohistochemistry assessed the effect of PPX on the quantity of NeuN-positive cells in the spinal cord tissue after SCI. Our findings revealed that the BBB and slanting board test scores of the PPX-treated model groups were considerably higher for the SCI group and significantly lower for the sham operation group ($P < 0.001$). Moreover, the PPX-2.0 group exhibited significantly higher NeuN expression levels than the SCI group ($P < 0.01$). Our findings indicate that PPX exerts a neuroprotective effect in secondary neuronal injury following SCI, facilitating the recovery of hind limb function by downregulating Bax/Bcl-2, caspase-3, and IL-1β.

## INTRODUCTION

Spinal cord injury (SCI) is a prevalent clinical disorder that causes persistent neurological deficits, including paraplegia and neuropathic pain, imposing substantial challenges for patients and their families (*Semita et al., 2023*). The high incidence, disability rate, and treatment costs associated with SCI place considerable economic burdens on their families and society (*Lo, Chan & Flynn, 2021*). Pathophysiologically, SCI is classified into primary and secondary injuries (*Bradbury & Burnside, 2019*). The primary injury is characterized by rapid and irreversible mechanical damage, often resulting in tissue destruction (*Anwar,*

Corresponding author
Lixin Zhu, zhulixin1966@163.com

*Al Shehabi & Eid, 2016*). Moreover, the primary injury triggers the release of various detrimental cytokines, causing changes in the microenvironment of the injured spinal cord and resulting in secondary injury (*Kopper & Gensel, 2018*). These secondary injuries could produce numerous inflammatory cytokines (such as tumor necrosis factor-α and interleukin-1β), accumulating within the damaged tissues, thereby forming a vicious cycle that aggravates the SCI. Unfortunately, current clinical therapies for SCI are ineffective in achieving satisfactory outcomes (*Yin et al., 2023*). Consequently, reducing secondary injury and protecting residual neurons are critical to treating SCI. Therefore, it is urgent to explore novel therapeutic strategies to address these challenges and advance the treatment of SCI.

Studies have demonstrated the potential of dopamine receptor D2 activators in alleviating oxidative damage following stroke (*Zhang et al., 2023*; *Pearson-Fuhrhop et al., 2013*), thereby identifying dopamine receptors as viable treatment targets for stroke. Pramipexole (PPX) is a non-ergot D2 type agonist that targets dopamine D2 receptors (DRD2) and dopamine D3 receptors (DRD3) (*Abdel et al., 2022*). The United States Food and Drug Administration has approved PPX as a clinical drug for treating Parkinson's disease and restless legs syndrome (*Pich & Collo, 2015*). Notably, PPX can reduce oxidative damage and mitochondrial dysfunction following brain injury (*Salman, Tabassum & Parvez, 2020*), exerting neuroprotective effects. Despite the observed neuroprotective effects in various conditions, the specific role of PPX in traumatic SCI remains largely unexplored. Dopamine receptors can be classified into D1 and D2 according to their biochemical and pharmacological properties. Class D1 receptors include D1 and D5 receptors, while class D2 receptor subtypes include D2, D3, and D4 receptors (*van Dijken et al., 1996*; *Yokoyama et al., 1994*). Studies have revealed diverse dopamine nerve dominations in the sensory and motor areas of the spinal cord in rats, cats, and monkeys, influencing various functions (*Holstege et al., 1996*). Interestingly, studies have indicated that DRD2 agonists have anti-inflammatory effects on the central nervous system in rat models of Pakinson's disease (*Jiang et al., 2022*). While few studies have demonstrated the presence of DRD2 receptors in the spinal cord tissue, the potential changes in DRD2 after the injury remain largely unknown. Therefore, we explored the neuroprotective effects of PPX on traumatic SCI and investigated the expression of DRD2 in post-SCI tissue.

Our findings revealed that PPX facilitates the recovery of lower limb function in rats with SCI and exerts neuroprotective effects by inhibiting apoptosis. Therefore, PPX is a promising therapeutic candidate for the treatment of SCI, offering the potential to enhance the efficacy of SCI.

## MATERIALS AND METHODS

### Chemicals and reagents

Pramipexole (HY-17355) was obtained from MedChemExpress (Monmouth Junction, NJ, USA). Nissl Staining Solution and ELISA Assay Kit were provided by Beyotime (Shanghai, China). The primary antibodies used in this study were anti-GADPH (ab9485; Abcam, Cambridge, UK), anti-DRD2 (DF10211; Affinity, Shanghai, China), anti-Bax (ab32503; Abcam, Cambridge, UK), anti-NeuN (ab196495; Abcam, Cambridge, UK), anti-Caspase-3

(BS1518; Bioworld, Nanjing, China) and anti-Bcl-2 (ab196495; Abcam, Cambridge, UK). Goat anti-rat IgG (HRP) (SA00001-2; Proteintech, Wuhan Shi, China) was used as the secondary antibody for western blotting.

## Animals

Studies involving adult female Sprague-Dawley rats were carried out following the "Guide for the Care and Use of Laboratory Animals". The research protocol received ethical approval from the Southern Medical University Animal Ethics Committee (approval ID: LAEC-2021-098). Adult female Sprague–Dawley rats weighing 180–200 g were procured from the Experimental Animal Center of Southern Medical University (China).
All animals were subsequently raised in a pathogen-free environment with regular light/dark cycles and provided *ad libitum* access to water and food. The rats were allowed to acclimatize for a week, following which they were randomly assigned to either the control or treatment group.

## Establishment of the SCI rat model and PPX treatment

One week after acclimatization, all animals were anaesthetized with an intraperitoneal injection of pentobarbital sodium solution in 0.9% saline (3%, 1.5 mL/kg) 30 min before the surgery. The rats were then secured onto a rat spinal cord adapter (68095; RWD, Guangdong, China). Laminectomy was performed at the tenth thoracic vertebra (T10) using a surgical microscope (SMZ745; Nikon, Minato City, Japan) and a grinding drill (South Korea 90 dental electric grinder), exposing the T10 spinal cord. The T10 spinal cord was fully exposed, revealing a smooth surface and a thick and visible vein in the middle of the back. The spinal cord was impacted by a precision impactor (speed 2.0 m/s, depth 1.5 mm dwell time 0.4 s, 68099II; RWD, Guangdong, China), causing blood vessel bleeding, as well as swelling and congestion of the spinal cord beneath the soft spinal cord, as observed with the surgical microscope. The incision was subsequently sutured in layers using 5-0 threads. After the SCI, the rats' bladders were manually drained twice daily until the bladder function was regained. The same group of researchers performed all the SCI operation procedures. After the surgery, all rats were administered with an intraperitoneal injection of 1 mL of saline containing $1 \times 10^5$ units of penicillin and 1 mg/kg meloxicam for 1 week to prevent infection and dehydration as well as relieve pain.

Female adult Sprague-Dawley rats were divided into four groups ($n = 24$). (1) Sham group: Rats in this group underwent laminectomy only and were administered 0.9% normal saline solution. (2) SCI group: Rats in this group underwent the SCI procedure described above and received intraperitoneal injections of 0.9% normal saline solution. (3) SCI+PPX (0.25 mg/kg) group: Rats in this group were intravenously injected with 0.25 mg/kg PPX in the tail daily for 28 consecutive days after injury. (4) SCI+PPX (2.0 mg/kg) group: Rats in this group were intravenously injected with 2.0 mg/kg PPX in the tail daily for 28 consecutive days after injury. Pramipexole (HY-17355; MedChemExpress, Monmouth Junction, NJ, USA) was prepared in normal saline at 1 mg/ml.

## Basso, Beattie, Bresnahan (BBB) evaluation of locomotion

The movement of both hind limbs of the rats was scored on days 0, 1, 3, 7, 14, 21, and 28 post-injury by two observers blinded to the animal groupings. The BBB evaluation scale ranges from 0 (no observable movement) to 21 (normal locomotion). The rats were placed on a rubber-padded, rectangular hardwood inclined plate with a 5° slope to conduct the evaluations. The inclined plate was then rotated by 5° at a time, and the greatest angle at which the rats could maintain their position for 5 s was recorded. Each animal underwent five evaluations, with the average value as the representative measurement.

## RNA preparation and qRT-PCR

Total RNA was extracted from the spinal cord tissue using TRIzol (Invitrogen, Waltham, MA, USA) following the manufacturer's instructions. The quality of the isolated RNA was evaluated using a NanoDrop 2000 spectrophotometer (Thermo Fisher Scientific, Waltham, MA, USA). The extracted RNA was then reverse transcribed into cDNA using the SuperScript$^{TM}$ kit (#10928–034; Invitrogen, Waltham, MA, USA). SYBR Green Pro Taq HS Supermix (AG11701; Accurate Biotechnology, Changsha, China) was used to quantify the mRNA expression level using the Real-Time PCR Detection System (Applied Biosystems in the United States). Each sample underwent at least three independent qRT-PCR amplifications to ensure reliability. Relative gene expression was estimated using the $2^{-\Delta\Delta Ct}$ method. The primers used for amplification are listed in Table 1.

## Western blotting analysis

The spinal cord tissues excised from the experimental animals were lysed in ice-cold RIPA lysis buffer (P0013B; Beyotime, Shanghai, China) supplemented with 100 mM phenylmethylsulfonyl fluoride (PMSF). The tissue was incubated in the lysis buffer at 4 °C for 60 min, followed by centrifugation to extract the supernatant. The protein concentration in the supernatant was determined using the BCA kit (CWBIO, Beijing, China) following the manufacturer's instructions. The samples were mixed with 5 × loading buffer and heated at 100 °C for 10 min before electrophoresis. The samples were separated on a 12% sodium dodecyl sulfate-polyacrylamide gel (EpiZyme, Shanghai, China) and transferred to nitrocellulose membranes (Millipore, Burlington, MA, USA). The membranes were blocked with 5% skimmed milk for 1 h to prevent non-specific binding. After blocking, the membranes were incubated overnight at 4 °C with specific primary antibodies (bax, bcl-2, caspase-3, DRD2, NeuN, GAPDH) at a dilution ratio of 1:1,000. The next day, the membranes were washed thrice with tris-buffered saline containing 0.1% Tween® 20 detergent (TBST) to remove unbound primary antibodies. Subsequently, the membranes were incubated with appropriate secondary antibodies (goat anti-rabbit, dilution ratio 1:10,000) for 1 h. The ECL detection kit (Bioship, Shanghai, China) was used to visualize the protein bands on the western blot. Subsequently, the images were analyzed using Quantity One software (Bio-Rad, Hercules, CA, USA).

**Table 1  Primer sequences.**

| Name | Primer | Sequence |
|---|---|---|
| Rat GAPDH | Forward | GAAGGTCGGTGTGAACGGAT |
| | Reverse | CCCATTTGATGTTAGCGGGAT |
| Rat bax | Forward | CGTCTGCGGGGGAGTCAC |
| | Reverse | AGCCATCCTCTCTGCTCGAT |
| Rat bcl-2 | Forward | GGATTGTGGCCTTCTTTGAGTTC |
| | Reverse | CTTCAGAGACAGCCAGGAGAAAT |
| Rat IL-1β | Forward | AGAATGGGCAGTCTCCAGGG |
| | Reverse | GACCAGAATGTGCCACGGTT |
| Rat TNF-α | Forward | GACCCTCACACTCAGATCATCTT |
| | Reverse | CCTTGAAGAGAACCCTGGGAGTAG |
| Rat DRD2 | Forward | TCGAGCTTTCAGAGCCAACC |
| | Reverse | GCATCCATTCTCCGCCTGT |

## Enzyme-linked immunosorbent assay (ELISA)

The frozen spinal cord tissue was rinsed with pre-cooled 0.1 mol/L phosphate-buffered saline (PBS) to remove residual contaminants. The required tissue sample was weighed, chopped, homogenized (10 mg tissue + 10 uL PBS solution), and centrifuged for 20 min (2,000–3,000 rpm) to obtain a clear supernatant containing the target analyte. The ELISA assay was performed according to the manufacturer's instructions. The plate (BIO-TEK ELX808; BIO-TEK, El Segundo, CA, USA) was read at the detection wavelength of 450 nm.

## Hematoxylin-eosin (H&E) staining and Nissl staining

The rats were anesthetized with 3% pentobarbital sodium, fixed on the operating table, and infused with 200 mL of 0.1 mol/L of PBS and 200 mL of 4% paraformaldehyde (PFA) through the left ventricle. The spinal cord was carefully excised under a microscope and immersed in 4% PFA for 24 h. After fixation, the tissue was dehydrated in an alcohol gradient and then treated with different concentrations of xylene. The tissue samples were embedded into paraffin blocks and sliced into 4 μm thin sections using a paraffin sectioning machine (Thermo Fisher Scientific, Waltham, MA, USA).

For Hematoxylin-eosin (H&E) staining, the sections were stained with H&E (Servicebio, Wuhan, China), and the results were observed under a Nikon light microscope (E100; Eclipse, Ottawa, Canada). For Nissl staining, the paraffin sections of tissues were incubated in Nissl Stain at 60 °C for 30 min. After incubation, sections were differentiated with 95% alcohol, dehydrated, and sealed with neutral gum. The surviving neurons in the spinal cord were quantified in three non-overlapping visual fields at 400 × magnification.

## Immunohistochemical assay

The spinal cord was carefully removed under a microscope and immersed in 4% PFA for 24 h. After fixation, the tissue was sliced into 4 μm thin sections using a paraffin sectioning machine (Thermo Fisher Scientific, Waltham, MA, USA). The tissue sections were placed

in citric acid antigen repair buffer and subjected to microwave heating to facilitate antigen retrieval. The tissue sections were then placed in a 3% hydrogen peroxide solution and incubated in the dark at room temperature for 25 min to block endogenous peroxidase activity. Subsequently, the sections were blocked with 3% bovine serum albumin for 30 min at room temperature. The slices were then incubated with specific primary antibodies (Bcl-2, Bax, and DRD2 at 1:200, NeuN at 1:1,000) overnight at 4 °C. The next day, the tissue sections were incubated with secondary antibodies (horseradish peroxidase-labeled) at room temperature for 50 min. Finally, the tissue sections were stained with 3,3′-Diaminobenzidine, with a brown-yellow color indicating positive results.

### Statistical analysis

ImageJ and Prism Version 8.0.2 (GraphPad, San Diego, CA, USA) were used for image processing, while Photoshop (Adobe CS6) was employed to compile the figures. All data are presented as mean ± standard deviation from at least three independent experiments. One-Way ANOVA and Tukey's multiple comparisons were used for inter-group comparisons, and the least significant difference test was adopted for pairwise comparisons. A $P$-value $< 0.05$ was considered statistically significant.

## RESULTS

### PPX attenuates histological deficits and promotes neurobehavioral functional recovery post-SCI in rats

To assess the neurobehavioral function following SCI, we obtained the BBB and slanting board test scores on days 0, 1, 3, 7, 14, 21, and 28. All model groups exhibited significantly lower scores in both tests compared to the sham group.

At 1 and 3 days after SCI, no significant difference in BBB scores was observed between the groups. However, at 7 days after SCI, the BBB scores of the PPX-2.0 group were significantly higher than that of the SCI group (Fig. 1A). Similarly, at 21 days after SCI, the BBB scores of the PPX-0.25 group were significantly higher than that of the SCI group. No significant differences were observed in the slanting board test scores between the groups during the first 7 days after SCI. Nonetheless, at 14 days post-SCI, the slanting board test scores in the PPX-2.0 group were significantly higher than those in the SCI group (Fig. 1B). Furthermore, at 21 days post-SCI, the slanting board test scores of the PPX-0.25 group were significantly higher than those of the SCI group. These findings suggest the potential of PPX in improving hind limb function recovery.

Furthermore, histological evaluations were conducted through HE and Nissl staining of spinal cord tissues. The sham group exhibited a distinct demarcation between white and gray matter under the microscope, with normal neuron shape, cell density, nucleosome staining, and cytoplasm, and only a minor degree of vacuolar alterations. In contrast, all model groups exhibited a significant decrease in the number of normal neurons 3 days after SCI compared to the sham group, with apparent swelling, pyknosis, and vacuolation in neurons, as well as intense cytoplasmic staining (Fig. 1C). However, by 28 days post-SCI, the pathological changes in the PPX-2.0 group were significantly reduced compared to the SCI group, with increased numbers of normal neurons exhibiting clear nucleosomes,

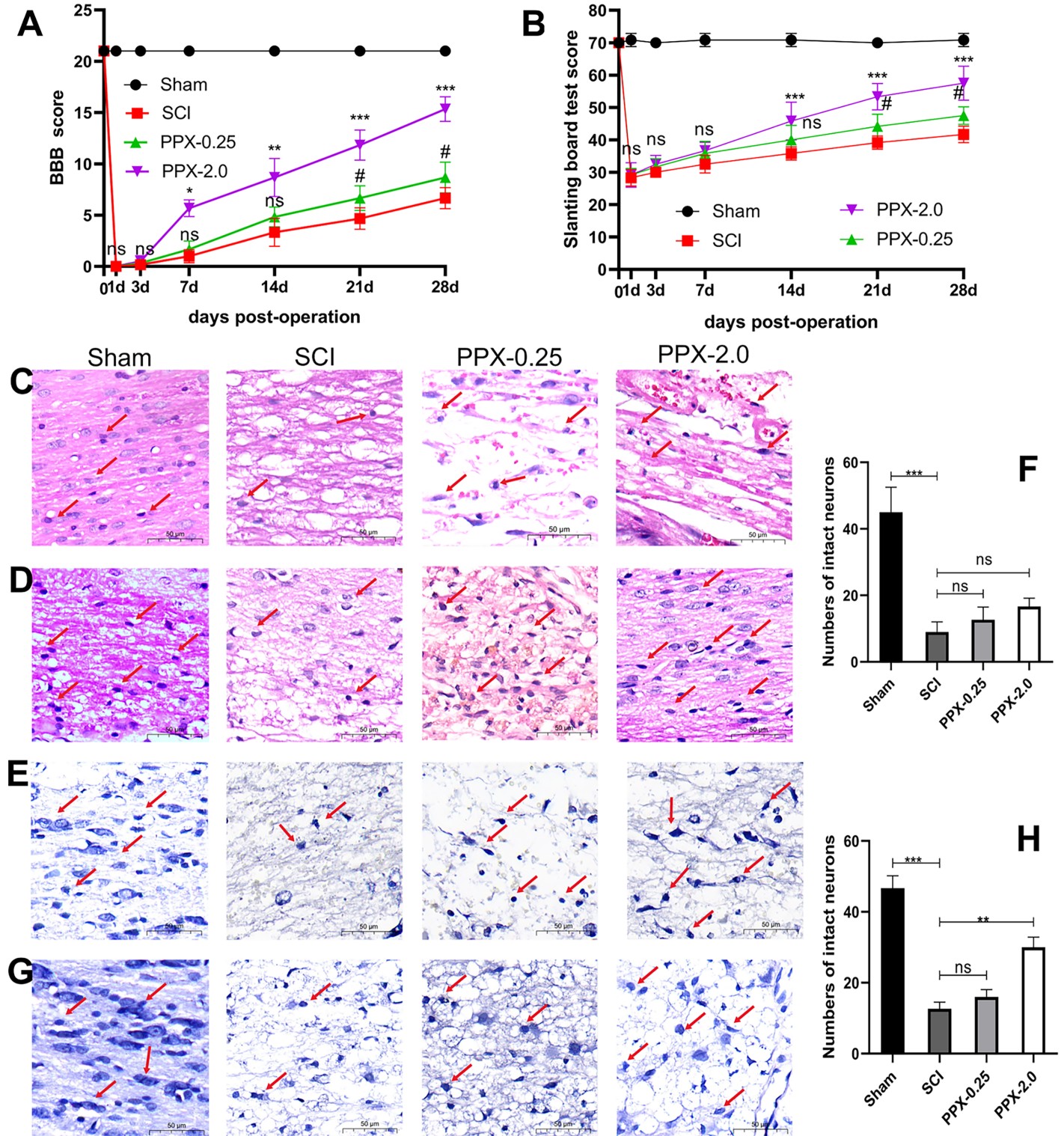

**Figure 1 PPX attenuated histological deficits and promoted neurobehavioral functional recovery after SCI in rats.** (A) Comparison of BBB scores at different times post-SCI. (B) Slanting board tests were compared at different times post-SCI ($n = 6$). Spinal cord tissue damage in SCI rats (×400, scale = 50 μm, $n = 3$). (C) HE staining of the spinal cord 3 days post-SCI. (D) HE staining of the spinal cord 28 days after SCI. (E) Nissl staining of the spinal cord 3 days post-SCI. (G) Nissl staining of the spinal cord 28 days post-SCI. (F and H) Quantitative analysis of the number of intact neurons 3 days (F) and 28 days (H) post-SCI. #$P < 0.05$ (SCI vs PPX-0.25), *$P < 0.05$, **$P < 0.01$, ***$P < 0.001$ (SCI vs PPX-2.0).

multipolar morphology, and few vacuolar changes (Fig. 1D). Additionally, the SCI group exhibited black and atrophic Nissl bodies 3 days post-SCI (Figs. 1E and 1F). However, in the PPX-0.25 and PPX-2.0 groups, the number of intact neurons increased, though without statistical significance at 3 days post-SCI (Figs. 1E and 1F). Nissl staining indicated that the number of intact neurons in the ventral horn increased in the PPX-2.0 group at 28 days post-SCI compared to the SCI group. On the other hand, the PPX-0.25 group did not exhibit statistical differences at 28 days post-SCI (Figs. 1G and 1H). These results suggest that PPX effectively attenuates histological deficits associated with SCI in rats.

## PPX inhibits apoptosis in the spinal cord tissue post-SCI in rats

To study the effect of PPX on apoptosis in injured spinal cord tissue, we employed western blotting to measure the expression levels of Bax/Bcl-2 and caspase-3 proteins in the spinal cord tissue 28 days post-SCI. The SCI group exhibited higher levels of Bax/Bcl-2 and caspase-3 protein expression compared to the sham group, while they were lower in the PPX-0.25 and PPX-2.0 groups compared to the SCI group (Figs. 2A–2C). The mRNA levels of Bax and Bcl-2 in the spinal cord tissue were determined using quantitative RT-PCR 7 days post-SCI. The mRNA levels of pro-apoptotic factor Bax had a greater relative expression in the SCI group than in the sham group; however, Bax mRNA levels were significantly lower in the PPX-2.0 group than in the SCI group (Fig. 2D). Additionally, the relative mRNA level of the anti-apoptotic gene Bcl-2 was higher in both the PPX-0.25 and PPX-2.0 groups compared to the SCI group, while the SCI group exhibited lower Bcl-2 expression compared to the sham group (Fig. 2E). We further employed immunohistochemistry to analyze the expression levels of Bax and Bcl-2 in the spinal cord tissue at various time points after injury in both the SCI and PPX-treated groups. To quantify the expression levels of Bax and Bcl-2, we used a computerized image analysis system to determine the mean optical density (MOD), which correlated with the positive staining intensity of Bax and Bcl-2. We found that the MOD of Bax expression in the spinal cord tissue was significantly higher in the SCI group at 3 days post-SCI compared to the sham group (Figs. 2F and 2H), whereas the MOD of Bcl-2 expression was lower in the SCI group than in the sham group (Figs. 2G and 2I). These findings are consistent with an increase in pro-apoptotic factors and a decrease in anti-apoptotic factors. Notably, at 7 days post-SCI, the PPX-2.0 group exhibited significantly lower MOD values for Bax-positive cells than the SCI group. However, there was no significant difference in Bax-positive cell MOD values between the PPX-0.25 and SCI groups (Figs. 2J and 2L). Furthermore, no significant difference was observed between the PPX-0.25 and SCI groups in the MOD values of Bcl-2-positive cells at 7 days post-SCI (Figs. 2K and 2M). These results collectively suggest that PPX may decrease neuronal apoptosis following SCI and provide neuroprotective effects through the regulation of Bax/Bcl-2 and caspase-3 levels, key regulators of the apoptotic pathway. These findings support the potential of PPX as a therapeutic agent for the treatment of SCI.

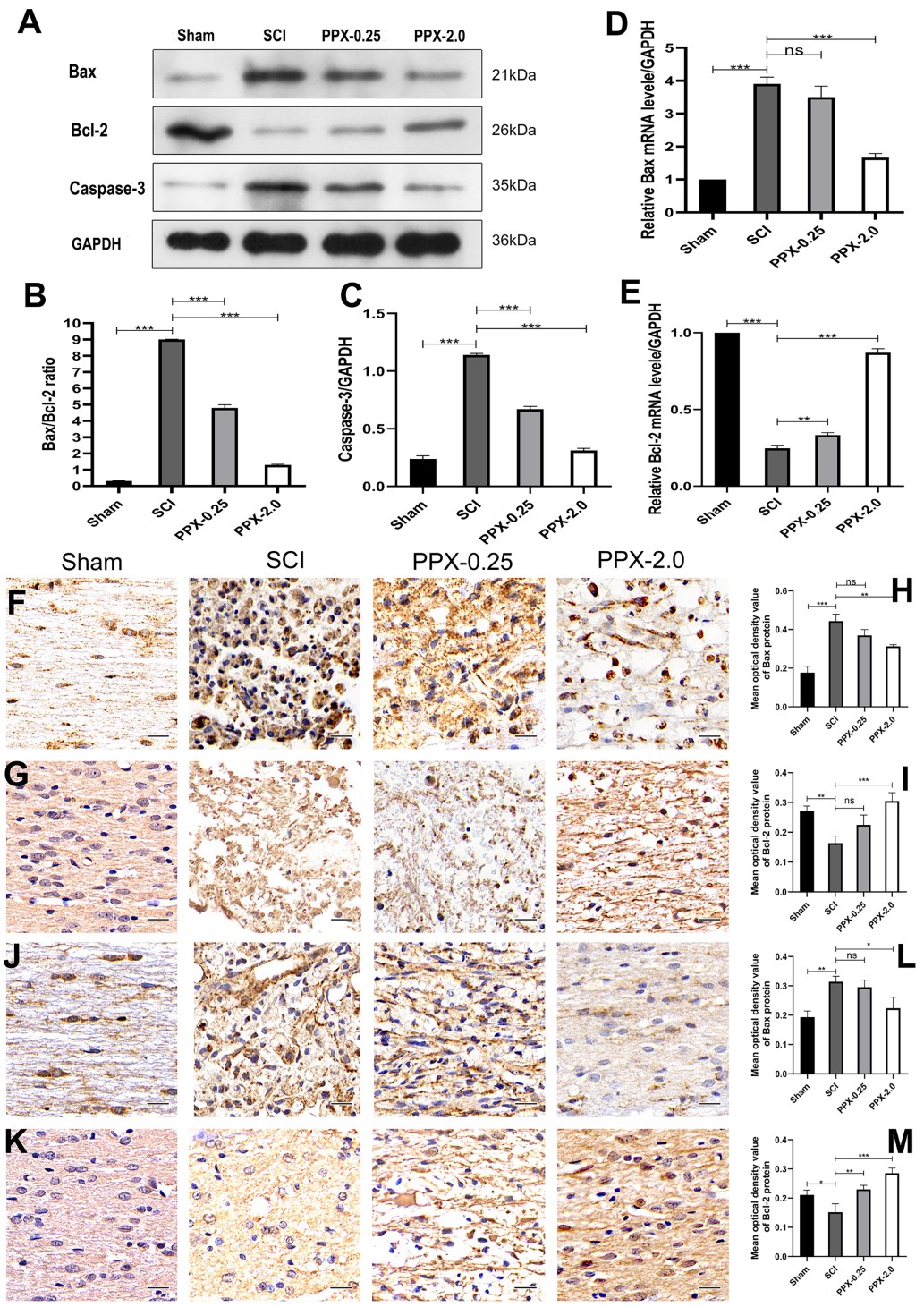

**Figure 2 Pramipexole inhibits apoptosis in spinal cord tissue post-SCI in rats.** (A) Western blot analysis of Bax, Bcl-2, and Caspase-3 expression level in the spinal cord tissue 28 days post-SCI. (B and C) Quantitative analysis of Bax/Bcl-2 (B) and Caspase-3 (C) protein levels. (D and E) qRT-PCR analysis showing the relative mRNA levels of Bax (D) and Bcl-2 (E) in spinal cords from Sham, SCI, PPX-0.25, and PPX-2.0, 7 days post-SCI. (F) The immunohistochemical staining of Bax in the spinal cord tissue, 3 days post-SCI. (G) The immunohistochemical staining of Bcl-2 in the spinal cord tissue, 3 days post-SCI. (H and I) Quantitative analysis of Bax-positive (H) and Bcl-2-positive (I) cells. (J and K) The

**Figure 2** (continued)
immunohistochemical staining of Bax (J) and Bcl-2 (K) in the spinal cord tissue, 7 days post-SCI. (L and M) Quantitative analysis of Bax-positive (L) and Bcl-2-positive (M) cells (×400, scale = 20 μm, $n = 3$). $^*P < 0.05$, $^{**}P < 0.01$, $^{***}P < 0.001$.             

## PPX inhibits inflammation in the spinal cord tissue after SCI in rats

To elucidate the anti-inflammatory effects of PPX post-SCI, we used ELISA to measure the levels of tumor necrosis factor-α (TNF-α) and interleukin-1 (IL-1) in the spinal cord tissue at 1, 3, and 7 days post-SCI. The SCI group exhibited higher IL-1β and TNF-α protein concentrations than the sham group. In contrast, the PPX-0.25 and PPX-2.0 groups demonstrated lower levels of IL-1β and TNF-α than the SCI group (Figs. 3A and 3B). We also assessed the relative mRNA expression levels of IL-1β and TNF-α in the spinal cord tissue 7 days post-SCI using quantitative RT-PCR. Consistent with the ELISA results, we observed higher relative mRNA expression of IL-1β and TNF-α in the SCI group than in the sham group. Conversely, the PPX-0.25 and PPX-2.0 groups displayed lower relative mRNA expression levels of IL-1β and TNF-α compared to the SCI group (Figs. 3C and 3D). These findings suggest that PPX treatment effectively inhibits inflammation in the spinal cord tissue after SCI.

## PPX increases NeuN and DRD2 expression after SCI in rats

Although PPX has demonstrated efficacy in improving hind limb function and promoting recovery after SCI, the underlying mechanism remains elusive. We investigated the change in the expression of DRD2, a protein induced by PPX, in the spinal cord tissue after SCI and PPX administration. Our findings revealed a lower expression of DRD2 in the spinal cord tissue of the SCI group compared to the control group (Figs. 4A and 4B). To assess the impact of PPX on DRD2 expression, we employed western blotting to measure DRD2 and NeuN protein levels in the spinal cord tissue 28 days after SCI. Notably, the PPX-0.25 and PPX-2.0 groups exhibited substantially higher DRD2 expression than the SCI group (Figs. 4C and 4D). While NeuN expression levels between the PPX-0.25 and SCI groups did not differ substantially, the PPX-2.0 group displayed a significantly greater NeuN expression compared to the SCI group (Figs. 4C and 4E). To further validate these findings, we utilized quantitative RT-PCR to analyze the relative expression of DRD2 mRNA in the spinal cord tissue 7 days post-SCI. Although DRD2 mRNA levels exhibited no statistically significant difference between the SCI and PPX-0.25 groups, it was considerably greater in the PPX-2.0 group than in the SCI group (Fig. 4F). Additionally, we observed lower relative expression of DRD2 mRNA in the SCI group than in the sham group. The impact of PPX on NeuN-positive cells in the spinal cord tissue 28 days post-SCI was assessed by immunohistochemistry. Consistent with the western blot findings, although there was no significant difference between the SCI and PPX-0.25 groups, the number of NeuN-labeled cells was significantly higher in the PPX-2.0 group than in the SCI group (Figs. 4G and 4H). Furthermore, we found a significantly lower number of NeuN-positive cells in the spinal cord tissue of the SCI group compared to the sham group. Therefore, our findings

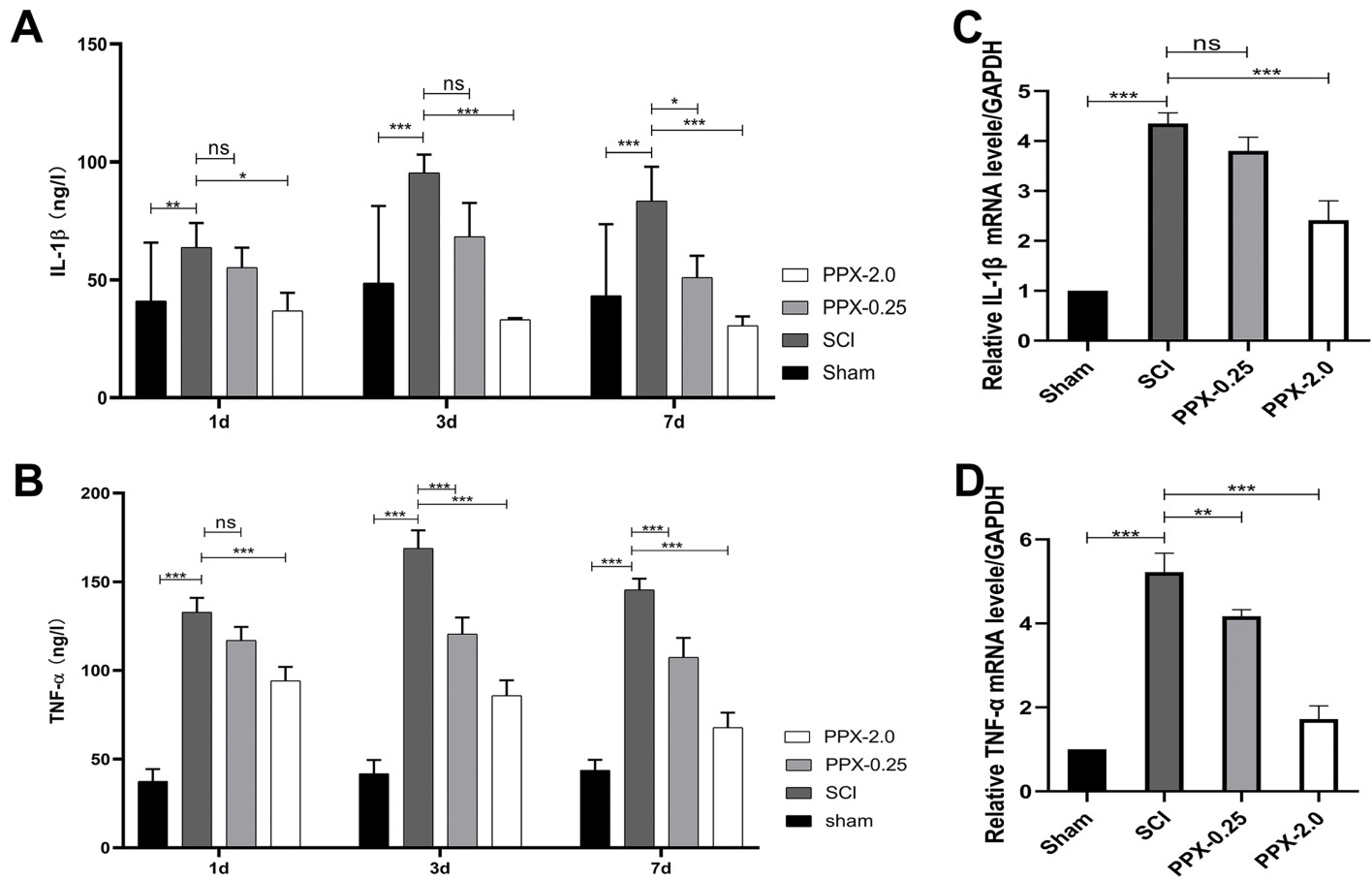

**Figure 3 Pramipexole inhibits the inflammation in spinal cord tissue post-SCI in rats.** (A and B) ELISA test for the concentration of inflammatory factors IL-1β (A) and TNF-α (B) at different times post-SCI. (C and D) RT-qPCR for mRNA of inflammatory factor genes IL-1β (C) and TNF-α (D) ($n = 3$). $^*P < 0.05$, $^{**}P < 0.01$, $^{***}P < 0.001$; ns, not significant.   

demonstrate that PPX treatment could lead to an increase in NeuN expression, which is associated with an increase in DRD2 expression in the spinal cord tissue following SCI.

## DISCUSSION

Spinal cord injury is a devastating condition characterized by two distinct stages: primary and secondary injury (*Semita et al., 2023*). The primary injury is typically irreversible (*Anwar, Al Shehabi & Eid, 2016*) and involves the initial mechanical damage to the spinal cord, often caused by trauma. On the other hand, the secondary injury is a complex cascade of pathological events that occur following the primary injury (*Kopper & Gensel, 2018*). Secondary injury includes cell death, edema, glutamate excitatory toxicity, blood-spinal cord barrier permeability, ion imbalance, inflammation, and lipid peroxidation (*Ni et al., 2023*). Although the secondary injury is potentially reversible, it significantly contributes to the poor clinical outcomes observed in patients with SCI. Therefore, it is necessary to identify new therapeutic strategies to slow down secondary SCI. This study found that PPX improved motor function and limb muscle strength in rats

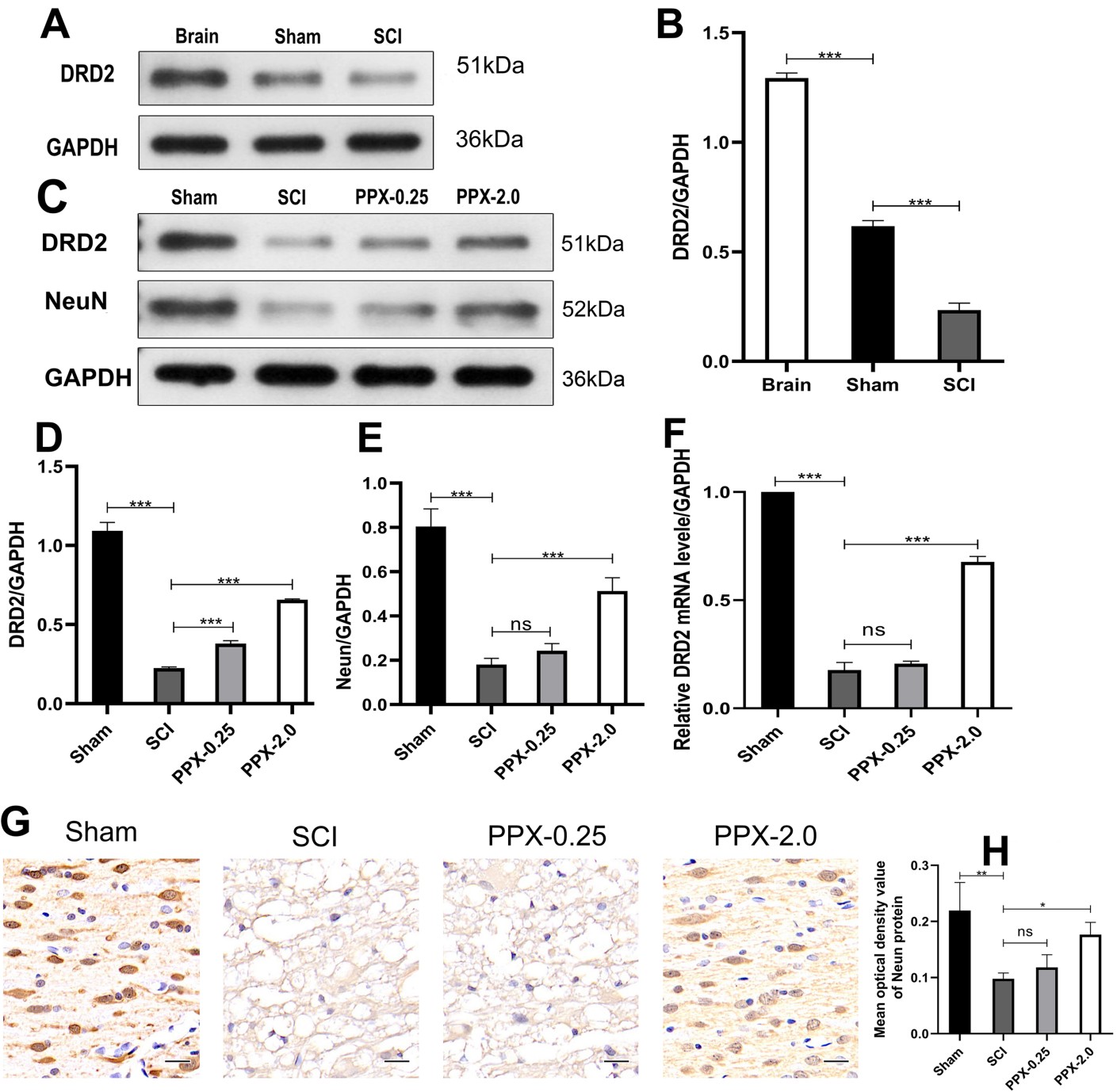

**Figure 4 Pramipexole increases the expression of NeuN and DRD2 post-SCI in rats.** (A) Western blot analysis of DRD2 expression level in the Brain, Sham, and SCI. (B) Quantitative analysis of DRD2 protein levels. (C) Western blot analysis of DRD2 and NeuN expression level in the spinal cord tissue at 28 days post-SCI. (D and E) Quantitative analysis of DRD2 (D) and NeuN (E) protein levels. (F) qRT-PCR analysis showing the relative mRNA level of DRD2 in spinal cords from Sham, SCI, PPX-0.25, and PPX-2.0 ($n = 3$). (G) The immunohistochemical staining showing NeuN expression in the spinal cord at 28 days post-SCI. (H) Quantitative analysis of NeuN-positive cells at 28 days post-SCI (×400, scale = 20 μm, $n = 3$). $^{*}P < 0.05$, $^{**}P < 0.01$, $^{***}P < 0.001$.

with SCI. Additionally, PPX also inhibited apoptosis and reduced the inflammatory response of the injured spinal cord tissue.

The clinical outcome of SCI treatment is still poor (*Yin et al., 2023*). Emergency surgery is frequently performed for spinal cord decompression and to maintain vertebral stability to prevent reinjury caused by vertebral dislocation. Additionally, frequent use of high doses of Methyl-prednisolone (MP) treatment improves the local absence of blood in SCI tissues and controls inflammation (*Chen et al., 2020*; *Wu et al., 2022*). However, these treatments are less effective and have limitations, leading to postoperative complications, making it crucial to identify new strategies to improve the treatment effect of SCI. In recent years, the dopamine D3/2 receptor stimulant, PPX, has been approved by the U.S. Food and Drug Administration for clinical use in the treatment of Parkinson's disease and restless legs syndrome (*Pich & Collo, 2015*). Studies have shown that PPX can reduce oxidative damage, mitochondrial dysfunction, and apoptosis, thereby playing a neuroprotective role (*Rasheed, Tabassum & Parvez, 2017*). Consistent with this, we also found that PPX can reduce the levels of pro-apoptotic proteins Bax and Caspase-3, along with an increase in the levels of the anti-apoptotic protein Bcl-2. Moreover, we also found that PPX can reduce the levels of local inflammatory cytokines IL-1β and TNF-α in SCI tissue, suggesting its neuroprotective role. Previous studies have reported that PPX can relieve oxidative damage through the Nrf2/HO-1 pathway (*Abdel et al., 2022*; *Salman, Tabassum & Parvez, 2020*), thus playing a protective role for neurons after traumatic brain injury. It is also consistent with the result of a study that pramipexole can reduce the damage to neurons in the central nervous system (*Dong et al., 2023*). These findings suggest that pramipexole may serve as a promising candidate as a potential treatment for SCI.

Dopamine (DA) is a crucial neurotransmitter in the brain that mediates a variety of functions, including motor activity, cognition, mood, positive enhancement, food intake, and endocrine regulation (*Wang et al., 2020*). Dopamine nerve fibers in the spinal cord are primarily dominated by the brain. The dopamine receptors belong to the G-protein coupling receptor and could be divided into two subfamilies: D1-like and D2-like, according to their pharmacological and biochemical properties. Both dopamine receptors are widely distributed in brain tissue. Work by *Levant & McCarson (2001)* has shown the presence of both D1 and D2 receptors in the superficial layers of the dorsal horn of the spinal cord. Dopamine D2/D3 receptors are present throughout the spinal cord and can be measured using the PET imaging agent 18 F-fallypride (*Yang et al., 2015*). Our study confirmed the presence of DRD2 protein in both brain tissue and thoracic spinal cord through western blotting, with lower expression levels in the thoracic spinal cord compared to the brain tissue, consistent with previous findings (Figs. 4A and 4B). Additionally, we observed a significant decrease in the expression of DRD2 post-SCI (Figs. 4A and 4B), and PPX treatment led to a significant increase in the level of DRD2 in the SCI tissue. UNC9995, a biased DRD2 receptor agonist, has been shown to reduce the secretion and suppression of inflammatory factor INF-β (*Zhu et al., 2020*), thereby slowing down the neurodegeneration of Parkinson's disease. Based on these findings, we propose that PPX could inhibit apoptosis in the damaged spinal cord tissue through DRD2 receptors and

suppress the local inflammatory response, leading to increased neuronal survival and improving motor function in rats after SCI.

Spinal cord injury leads to the activation of surviving microglia and vascular endothelial cells, releasing inflammatory cytokines and chemokines, such as TNF-α, IL-1β, and MCP-1 (*Boato et al., 2013*). Studies have shown that acrylamide, combined with titanium dioxide nanoparticles, can reduce TNF-α and IL-1β levels, inhibiting neuronal death (*Safwat, Mohamed & Mohamed, 2022*). TNF-α has been implicated in the death of spinal motor neurons and the exacerbation of cell death after SCI. Local application of anti-TNF-α serum can reduce spinal edema, microvascular permeability, and cellular damage in SCI rats, while reducing IL-1β can prevent injury development and enhance axis plasticity (*Boato et al., 2013*), promoting neural function restoration. Therefore, post-SCI regulation of inflammatory factors such as TNF-α and IL-1β can protect early residual neurons, creating prerequisites for subsequent spinal cord regeneration. We also observed an increase in inflammatory cell factors in spinal cord tissue after SCI. However, PPX significantly reduced the levels of IL-1β and TNF-α. Additionally, apoptosis of neurons often occurs within 6 h to 3 weeks following SCI (*Allison & Ditor, 2015*), resulting in further damage to spinal cord tissue. Apoptosis, a process of programmed cell death, proceeds through the activation of an evolutionary conserved intracellular pathway that is essential in the homeostasis of normal tissues. Apoptosis of neurons is key in secondary injury after SCI (*Liu et al., 2023*). Previous studies have demonstrated that the accumulation of cytokines can activate astrocytes and ultimately lead to neuronal apoptosis (*Simi et al., 2007*; *Allan, Tyrrell & Rothwell, 2005*). In this study, we found a significant increase in the levels of pro-apoptotic proteins Bax and Caspase-3 in the injured area after SCI, while the level of the anti-apoptotic protein Bcl-2 significantly decreased. Importantly, PPX effectively reduced the levels of Bax and Caspase-3 while increasing the expression of the anti-apoptotic protein Bcl-2, suggesting its potential protective role by inhibiting apoptosis post-SCI. The observed neuroprotective effects of PPX offer promising potential as a treatment strategy for spinal cord injury. However, it is essential to acknowledge the limitations of our study. Firstly, although we identified that PPX exerts neuroprotective effects by inhibiting apoptosis and inflammation in SCI in rats, the specific underlying mechanisms through which PPX achieves these effects require further investigation and should be explored in future studies. Furthermore, while our study demonstrated that PPX can upregulate the expression of DRD2 in injured spinal cord tissue, the precise regulatory mechanism by which PPX achieves this remains unclear and warrants further exploration.

## CONCLUSION

Our study demonstrated that PPX promotes the recovery of nerve function following SCI in rats. We also observed that PPX effectively suppresses apoptosis and the local inflammatory response in the injured area after SCI in rats. Additionally, we found that DRD2 was expressed in thoracic spinal cord tissue, and the expression level of DRD2 decreased after spinal cord injury. Furthermore, PPX can upregulate DRD2 expression in injured spinal cord tissue. These findings highlight the potential efficacy of PPX as a

therapeutic agent for treating SCI, offering a promising and novel approach to improve outcomes for patients with SCI.

## ACKNOWLEDGEMENTS

The authors would like to thank the Department of Spinal Surgery, Orthopedic Medical Center, Zhujiang Hospital, and Southern Medical University for providing the laboratory facilities and administrative support to conduct the study.

### Funding

This work was supported by grants from the Natural Science Foundation of China (No. 81974329 and No. 81672140). The funders had no role in study design, data collection and analysis, decision to publish, or preparation of the manuscript.

### Grant Disclosures

The following grant information was disclosed by the authors:
Natural Science Foundation of China: 81974329, 81672140.

### Competing Interests

The authors declare that they have no competing interests.

### Author Contributions

- Xuchen Liu conceived and designed the experiments, analyzed the data, prepared figures and/or tables, authored or reviewed drafts of the article, and approved the final draft.
- Chengqiang Wang conceived and designed the experiments, analyzed the data, prepared figures and/or tables, authored or reviewed drafts of the article, and approved the final draft.
- Qingshan Peng performed the experiments, analyzed the data, prepared figures and/or tables, and approved the final draft.
- Birong Peng performed the experiments, analyzed the data, prepared figures and/or tables, and approved the final draft.
- Lixin Zhu conceived and designed the experiments, authored or reviewed drafts of the article, and approved the final draft.

### Animal Ethics

The following information was supplied relating to ethical approvals (*i.e.*, approving body and any reference numbers):

The Ethics Committee of Southern Medical University.

### Data Availability

The raw data is available in the Supplemental File.

## Supplemental Information

Supplemental information for this article can be found online at http://dx.doi.org/10.7717/peerj.16039#supplemental-information.

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
