# Peer review of "Pramipexole has a neuroprotective effect in spinal cord injury and upregulates D2 receptor expression in the injured spinal cord tissue in rats"

_PeerJ, doi:10.7717/peerj.16039_

## Round 0.1 · original submission · Major Revisions

The two reviewers have put forward some very valuable comments on your manuscript. I hope you can reply and revise one by one according to these comments to improve the quality of the entire manuscript.

Reviewer 1 ·

Basic reporting

1.The study suggests that PPX has a neuroprotective effect in secondary neuron injury after spinal cord injury. It can promote the recovery of hind limb function by inhibiting levels of Bax/Bcl-2 and caspase-3, which are involved in apoptosis, and potentially inhibiting the expression of IL-1β, a pro-inflammatory cytokine. This finding implies that PPX may modulate dopamine receptor signaling, which could contribute to its neuroprotective properties. However,grammar issues such as spelling errors, incorrect use of tenses, and inconsistent subject predicate in the article need to be checked and corrected by a proper reviewing service.

2.In general, the reference situation is basically reasonable. In order to better propose the objectives of this study, the author needs to describe the current mechanism of SCI and the possible mechanisms of PPX in the Introduction.

3.The chapters of the article are basically reasonable, and the conclusion will be clearer if it is divided into separate paragraphs.
The drawing is basically professional, the scale is reasonable, the drawing style is simple and clear. But the font size and format in the Figures should be consistent, such as labeling in the Figures, the colour in Figure 2 and 4.
The raw data is complete with no obvious gaps and inconsistencies.

Experimental design

4.Please explain how PPX solutions were made in the materials and methods. What is the buffer for dissolving the drug? Were sham and SCI group administered with the buffer as control?
5.The product number of the reagent needs to be specified in detail, such as the dilution ratio of anti-Bax (ab32503 Abcam UK), anti-Bcl-2 (ab196495 Abcam UK), anti-caspase-3 (BS1518 Bioworld China), anti-DRD2; The ECL detection kit (Bioship, China); HE and Nissl tissue staining were performed according to the manufacturer’s instructions (Beyotime China).

Validity of the findings

6.Numerous methods were described in the results, such as “To study the effect of PPX on apoptosis in the injured spinal cord tissue, we measured the expression levels of Bax/Bcl-2 and caspase-3 proteins in the spinal cord tissue 28 days after SCI determined using western blotting; relative mRNA expression level of Bax and Bcl-2 in the spinal cord tissue 7 days after SCI determined using quantitative RT-PCR; and assessed the expression of pro-apoptotic factor Bax and anti-apoptotic factor Bcl-2 in spinal tissue 3 and 7 days after SCI using an immunohistochemistry technique.”, “We used ELISA to detect the levels of tumor necrosis factor (TNF) and interleukin-1 (IL-1) in the spinal cord tissue at 1, 3, and 7 days after SCI in order to research how PPX affects inflammation after SCI. We also assessed the relative mRNA expression level of IL-1β and TNF-α in the spinal cord tissue 7 days after SCI by quantitative RT‑PCR.”. Please reduce the description of the method in the results.
7.The effects of PPX-0.25 in inflammation in the spinal cord tissue after SCI in rats should be described in the Results.
8.The discussion part lacks the summary and induction of the previous article, and does not provide readers with a concise and clear conclusion, nor does it provide further thinking direction and research suggestions. It is suggested that the author should better clarify the thesis or main idea, so that the reader can understand the research value more clearly.

Reviewer 2 ·

Basic reporting

Liu et al. studied the neuroprotective effect of Pramipexole after SCI in rats. While the figures are well constructed, the text is difficult to understand. Extensive English language and style editing is needed to improve the manuscript's readability. Numerous grammar and punctuation mistakes were found. 

References need to be checked. In 284-286, the most abundant catecholamine in the brain is dopamine, which acts via dopamine receptors, which belong to G protein-coupled receptor superfamily that can be divided into D1-like and D2- like subfamilies...... the author should add some relative references to introduce dopamine receptors.

The data and figures are basically clear.

Experimental design

The author indicated that “According to our research, PPX causes an increase in NeuN expression, which is connected with an increase in DRD2 expression.” Based on the research design and current results, it cannot be proven that NeuN expression increase caused by PPX connected with DRD2 expression increase.

“We hypothesize that PPX can prevent apoptosis and the inflammatory response, which is mediated by DRD2 activation, and therefore increase the survival of neurons in the wounded area and safeguard the spinal cord tissue.” The article has already obtained clear results that PPX inhibited apoptosis and inflammation in spinal cord tissue after spinal cord injury (SCI) in rats. However, whether these processes mediated by DRD2 is unclear. Therefore, the description is not accurate. Please revise it.

Material and methods need to be further refined, including but not limited to reagent number manufacturers.

Validity of the findings

The arrowheads should be added in Fig 1C, D, E, G to indicate the positive results .

Additionally, 3 days after SCI, black and atrophic Nissl bodies were seen in the SCI group (Figure 1E), whereas Nissl staining and intact neuronal count in the ventral horn increased in the PPX-2.0 group compared to that in the SCI group at 28 days after SCI (Figure 4G). Figure 1F, G and H was not observed in the section of Results. Please confirm that whether Figure 4G is Figure 1G. In addition, the effects of PPX-0.25 in histological deficits and neurobehavioral functional recovery after SCI in rats should be described.

It is necessary to revise the discussion part, and the existing results should be discussed and analyzed, and the research conclusions should not be exaggerated.

Additional comments

It is necessary to explain the limitations of this study.

---

## Round 0.2 · accepted · Accept

All reviewers have accepted your response to their comments. Congratulations, your article is being considered for acceptance.

Reviewer 1 ·

Basic reporting

I think the author's revisions were meticulous and in place, basically addressing my concerns and some review comments.

Experimental design

no comment

Validity of the findings

no comment

Reviewer 2 ·

Basic reporting

The author responded well and revised my doubts, and this version can be approved.

Experimental design

no comment

Validity of the findings

no comment